# Functional Deficits in Gut Microbiome of Young and Middle-Aged Adults with Prediabetes Apparent in Metabolizing Bioactive (Poly)phenols

**DOI:** 10.3390/nu12113595

**Published:** 2020-11-23

**Authors:** Xuhuiqun Zhang, Anqi Zhao, Amandeep K. Sandhu, Indika Edirisinghe, Britt M. Burton-Freeman

**Affiliations:** Department of Food Science and Nutrition and Center for Nutrition Research, Institute for Food Safety and Health, Illinois Institute of Technology, Chicago, IL 60616, USA; xzhan198@iit.edu (X.Z.); azhao2@hawk.iit.edu (A.Z.); asandhu2@iit.edu (A.K.S.); iedirisi@iit.edu (I.E.)

**Keywords:** prediabetes, gut microbiome, (poly)phenolic metabolites, red raspberries, shotgun sequencing, UHPLC-QQQ

## Abstract

Background: Gut microbiota metabolize select dietary (poly)phenols to absorbable metabolites that exert biological effects important in metabolic health. Microbiota composition associated with health/disease status may affect its functional capacity to yield bioactive metabolites from dietary sources. Therefore, this study assessed gut microbiome composition and its related functional capacity to metabolize fruit (poly)phenols in individuals with prediabetes and insulin resistance (PreDM-IR, *n* = 26) compared to a metabolically healthy Reference group (*n* = 10). Methods: Shotgun sequencing was used to characterize gut microbiome composition. Targeted quantitative metabolomic analyses of plasma and urine collected over 24 h were used to assess microbial-derived metabolites in response to a (poly)phenol-rich raspberry test drink. Results: PreDM-IR compared to the Reference group: (1) enriched *Blautia obeum* and *Blautia wexlerae* and depleted *Bacteroides dorei* and *Coprococcus eutactus*. *Akkermansia muciniphila* and *Bacteroides* spp. were depleted in the lean PreDM-IR subset; and (2) impaired microbial catabolism of select (poly)phenols resulting in lower 3,8-dihydroxy-urolithin (urolithin A), phenyl-*γ*-valerolactones and various phenolic acids concentrations (*p* < 0.05). Controlling for obesity revealed relationships with microbial species that may serve as metagenomic markers of diabetes development and therapeutic targets. Conclusions: Data provide insight from multi-omics approaches to advance knowledge at the diet–gut–disease nexus serving as a platform for devising dietary strategies to improve metabolic health.

## 1. Introduction

Prediabetes (PreDM) is an intermediate stage in the development of Type 2 diabetes mellitus (T2DM) and presents as impaired fasting glucose (IFG) and/or impaired glucose tolerance (IGT). Globally, 373.9 million adults (7.5%) aged 20–79 years have IGT [1] and approximately 88 million (34.5%) qualify as having PreDM in the USA [2]. PreDM usually co-occurs with a cluster of cardio-metabolic risk factors that leads to substantially increased risk of developing T2DM and cardiovascular diseases [2]. Data suggest that the gut microbiome is associated with T2DM and PreDM risk factors [3,4,5,6,7]; however, much of this research is derived from older adults (60+ years old) [4,5,6]. The prevalence of PreDM in young adults (18–44 years old) is 28.7% in the USA [8], yet much less is known about the gut microbiome composition in younger adults with PreDM. In addition, individuals with PreDM typically present with obesity, which has been associated with an increased *Firmicutes/Bacteroides* ratio and reduced abundance of the genera *Bifidobacterium* and *Bacteroides* in the gut [9]. However, the composition of the gut microbiome in young and middle-aged adults with PreDM independent of obesity is not clear. 

Gut microbial composition affects host health by shaping the environment in the colon and producing metabolites that are involved in signaling and immune system modulation [10]. Microbial metabolite profiles are derived in part from the diet. Gut microbes catabolize unabsorbed dietary nutrients and other constituents, such as (poly)phenols, producing metabolites that influence local and systemic host physiology. Anthocyanins, ellagitannins and flavan-3-ols are example dietary (poly)phenols that rely greatly on microbial catabolism for absorption and subsequent biological effects [11,12]. Certain fruits like red raspberries, strawberries and pomegranates uniquely contain all three of these types of (poly)phenolic compounds and dietary intervention studies have demonstrated improved cardio-metabolic risk factors in an at-risk population [13,14,15], and reduced fat mass in animal models [16,17]. Though parent (poly)phenolic compounds have low bioavailability, clinical studies indicate extensive biotransformation to various bioactive metabolites, including urolithins, phenyl-*γ*-valerolactones and phenolic acids by the gut microbiota [11,18,19]. However, data are limited regarding the microbial capacity to catabolize (poly)phenols in disease and pre-disease states and whether associations with species-level gut microbes may serve as therapeutic targets for improving metabolic health through diet. 

Therefore, the main objectives of this research were: (1) to assess structural and functional characteristics of the gut microbiome in young and middle-aged adults with PreDM and insulin resistance (PreDM-IR) relative to a metabolically healthy reference control group (Reference); (2) to investigate the functional capacity of the gut microbiota in PreDM-IR compared to the Reference group in response to a (poly)phenol-rich test drink; and (3) to investigate the putative associations between species-level gut microbes, metabolic health status and microbial (poly)phenolic metabolites. 

## 2. Materials and Methods

### 2.1. Study Participants

This study was approved by the Institutional Review Board (Protocol #IRB2016-136) of Illinois Institute of Technology (IIT), Chicago, Illinois, and registered with ClinicalTrials.gov (NCT03049631). All subjects provided written informed consent before initiation of any study procedures. The clinical part of the study was conducted from May 2017 to February 2018 in the Center for Nutrition Research (CNR) at the IIT, Chicago, Illinois. 

Subjects were recruited from the greater Chicagoland area and were required to meet general and specific eligibility criteria for PreDM-IR. Specific PreDM-IR criteria included IFG (≥5.6 and <7.0 mmol/L), elevated fasting insulin (>50th percentile cutoff) [20] and IR measured by homeostatic model assessment of insulin resistance, HOMA-IR ≥ 2 [21]. The Reference group had fasting glucose values of <5.6 mmol/L and a HOMA-IR of ≤1 [21]. General eligibility criteria included: non-smoker and not taking any medications that would interfere with outcomes of the study, and no documented atherosclerotic, inflammatory, gastrointestinal or kidney diseases, diabetes mellitus or other systemic diseases. Women who were pregnant or lactating were not eligible to participate in the study. 

### 2.2. Study Design and Procedure

This study was an acute (24 h) multi-sampling design using multiple specimens (stool, urine and blood) and incorporating a dietary (poly)phenolic source (red raspberry-based test drink, RRBtest) to assess gut microbiome function to catabolize (poly)phenols relative to resident gut microbiota in two distinct groups representing metabolically impaired and healthy populations (Figure 1). Dietary habits were evaluated via the Automated Self-Administered 24-h (ASA24) Dietary Assessment Tool (version 2017, developed by the National Cancer Institute, Bethesda, MD, USA) [22]. A fecal sample was collected with a standard collection kit at home, kept in an insulated icebox and transported to the laboratory and stored at −80 °C within 24 h. 

After a 3-day low-(poly)phenol diet stabilization period, subjects participated in a postprandial day protocol that involved ingesting an RRBtest drink to assess the gut microbial capacity to generate microbial-derived (poly)phenolic metabolites. The RRBtest drink contained 250 g (~2 cups fresh weight equivalence) Individually Quick Frozen RRB (*Rubus idaeus* L. var. Wakefield, Enfield Farms, Lynden, WA, USA) (Table 1) [12]. On the postprandial day, subjects arrived (10–12 h fasted, well hydrated) in the morning. Anthropometrics, body composition and vital signs (blood pressure and heart rate) were measured. Body composition was measured with subjects wearing a light robe and no shoes using the Tanita Body Composition Analyzer Model BC-418 (Arlington Heights, IL, USA). A licensed health care professional placed an intravenous catheter in subjects’ non-dominant arm. Fasting blood and urine samples were collected (0 h) and subjects were provided with the RRBtest drink. Subsequently, blood and urine samples were collected at 1, 2, 3 and 4 h. After the 4-h blood collection, the catheter was removed, and subjects were evaluated for safety before leaving the CNR. Subjects consumed low-(poly)phenolic lunch and dinner meals at home and returned the next morning (10–12 h fasted, well hydrated) for the 24 h blood and urine collection to complete the postprandial intervention protocol.

### 2.3. DNA Extraction and Whole Genome Shotgun Sequencing

Fecal DNA extraction was carried out using the PowerFecal DNA isolation kit (QIAGEN, Hilden, Germany). Total DNA samples were cleaned with the Zymo DNA clean-up and concentrator kit (Zymo Research, CA, USA). Illumina sequencing libraries were prepared using the Nextera XT DNA Library prep kit and the Nextera XT Index Kit (Illumina, San Diego, CA, USA). Library concentrations were quantified with the Qubit dsDNA Assay Kit (Invitrogen, Carlsbad, CA, USA) and their sizes were estimated by electrophoresis with the Agilent High Sensitivity DNA Chip (Agilent Technologies, Santa Clara, CA, USA). Metagenomic libraries were sequenced on the Illumina MiSeq platform, targeting ~1.5 Gb of the sequence per sample with paired-end reads (2 × 250 bp).

Low-quality reads and bases in the sequence data generated from the fecal samples were filtered out with Trimmomatic (version 0.33) [23] and human contamination was removed by read mapping against the human genome with Bowtie2 (version 2.3.5) [24] using the KneadData pipeline [25]. Species-level taxonomic abundances were inferred for all samples using the FDA/CFSAN/OARSA/DMB in-house k-mer database (k = 30) [26]. The Human Microbiome Project Unified Metabolic Analysis Network (HUMAnN2 v.0.2.1) was used to investigate the presence/absence and the relative abundance of gene families and pathways in each sample to provide a functional interpretation of the metagenomic sequences [25]. Model-based Genomically Informed High-dimensional Predictor of Microbial Community Metabolic Profiles (MelonnPan) was used to predict community metabolomes from microbial community profiles [27].

### 2.4. Dietary Assessment

ASA24 was completed by subjects to record all food and drink consumed in the past 24 h. The total calorie, fat, protein, carbohydrate, sugar, fiber, vegetable and fruit amounts were calculated based on the ASA24 reports and analyzed with SAS 9.4 (SAS Institute, Cary, NC, USA).

### 2.5. Metabolic Health Indices Analysis

Blood samples were collected in vacutainers containing ethylenediaminetetraacetic acid (EDTA) and immediately placed on ice until being centrifuged (within 30 min). After centrifugation at 453× *g* for 15 min at 4 °C, plasma was aliquoted into individual Cryovials and stored at −80 °C until analysis. Spot urine samples were collected in urine collection cups, immediately placed on ice and then aliquoted into individual Cryovials and stored at −80 °C until analysis. Fasting plasma metabolic markers (glucose, insulin, total cholesterol (TC), high-density lipoprotein cholesterol (HDL-C) and triglycerides (TG)) and urine creatinine were assessed using an RX Daytona automated clinical analyzer (Randox Laboratories, Crumlin, UK) with appropriate reagents, standards and quality controls. Low-density lipoprotein cholesterol (LDL-C) was calculated using Friedewald’s equation [28]. Intra-assay variations were less than 3.2% and inter-assay variations were less than 5.5% (Appendix A). The atherogenic risk was estimated from TC/HDL-C, LDL-C/HDL-C and non-HDL-C/HDL-C (AI) [29]. Steady-state IR and β-cell function, i.e., HOMA-IR and HOMA-β, were calculated from fasting plasma glucose (FPG) and fasting plasma insulin (FPI) expressed in mmol/L and μIU/mL, respectively, with Equations (1) and (2) [30]:HOMA-IR = (FPG × FPI)/22.5(1)
HOMA-β = (20 × FPI)/(FPG − 3.5)(2)

### 2.6. Blood and Urine Microbial (Poly)phenolic Metabolites Analysis

The plasma and urinary analysis of (poly)phenolic metabolites was performed using an ultra-high performance liquid chromatography triple quadrupole (UHPLC-QQQ) model 6460 (Agilent Technologies) as previously described [12,31]. Plasma samples were extracted using solid-phase extraction (SPE) C18 cartridges (3 mL et al., 200 mg; Agilent Technologies). Urine samples were filtered with a 0.2 μm Polypropylene syringe filter (Whatman, Maidston, UK). Sample extracts (5 μL) were injected into a reversed-phase Poroshell C18 Stable Bond column (2.1 × 150 mm, 2.7 μm; Agilent Technologies) equipped with a guard column (2.1 × 5 mm, 2.7 μm; Agilent Technologies) for the separation of RRB (poly)phenols and their metabolites (except for phenolic acids). The mobile phases were 1% formic acid in water (A) and acetonitrile (B), with a gradient consisting of 5% B at 0 min, 15% B at 10 min, 20% B at 12 min, 50% B at 20 min and 90% B at 23 min followed by 7-min post-run time for column re-equilibration. A Pursuit 3 PFP column (2.0 × 150 mm, 3 μm; Agilent Technologies) equipped with a guard column (2.0 × 2 mm, 3 μm; Agilent Technologies) was used for the separation of phenolic acids and their derivatives. The mobile phases were 0.1% formic acid in water (A) and 0.1% formic acid in acetonitrile (B), with a gradient consisting of 5% B at 0 min, 10% B at 3 min, 15% B at 7–9 min, 20% B at 10–11 min, 25% B at 12 min, 30% B at 13–14 min and 95% B at 15 min followed by 5-min post-run time for column re-equilibration. Metabolite identification was performed by multiple reaction monitoring (MRM) optimized with pure standards wherever possible and parent/daughter ion fragments obtained from an UHPLC coupled with quadruple time-of-flight (UHPLC-QTOF) [12]. Method validation and MRM ion transitions for different metabolites have been reported in our recently published work [31]. Nomenclature of (poly)phenolic metabolites is based on recent recommendations [32]. Plasma (poly)phenol concentrations were expressed as nmol/L. Urinary (poly)phenol concentrations were adjusted for creatinine concentration and expressed as nmol/μmol creatinine. 

### 2.7. Statistical Analysis

Sample size estimates were performed using SAS 9.4. Thirty-one subjects allowed for > 80% power to detect > 70% difference in microbial metabolites after the RRBtest drink between PreDM-IR and Reference groups, assuming a 90% coefficient of variation (CV) between subjects. The sample size also allowed for > 80% power to detect > 75% difference in gut microbial abundance between PreDM-IR and Reference groups, assuming a 90% CV between subjects.

The within-sample diversity (α-diversity) was quantified by the Shannon index using the relative abundance profiles at species levels using Vegan (R version 3.5.3, Bioconductor package) [33]. The between-sample diversity (β-diversity) was quantified by principal component analysis (PCA) using Statistical Analysis of Taxonomic and Functional Profiles (STAMP) [34]. The biological difference and effect size estimation was calculated between groups (PreDM-IR and Reference) and subgroups (PreDM-Lean and Reference-Lean) using linear discriminant analysis effect size (LEfSe) with a cutoff value of a linear discriminant analysis (LDA) score (log_10_) above 2.0 and *p* < 0.05 [35].

Subgroup analyses to investigate possible microbial markers of PreDM-IR independent of obesity were based on criteria defining lean/healthy weight and overfat/obese groups according to the Tanita body fat percentage chart [36]. 

Plasma and urinary endpoints analyses were performed using SAS 9.4. The areas under the 24 h curve of (poly)phenolic metabolites (AUC_0–24h_) after the RRBtest drink were determined using the trapezoidal method [18]. Data distribution was examined via the Shapiro–Wilk test. Data not conforming to normal distribution patterns were log_10_-transformed. Data were analyzed using the MIXED procedure assessing main effects of groups (PreDM-IR vs. Reference) and subgroups (PreDM-Lean vs. Reference-Lean). Covariates (age, BMI, gender and race) were tested, and significant covariates were included in final analyses. When a significant main effect was observed, post hoc mean separation testing was conducted using the Tukey–Kramer correction to adjust for multiple comparisons. Spearman’s rank coefficient correlation analysis was used for correlations between metagenomic abundance at the species level and metabolic health indices and microbial metabolites (SAS 9.4). Subjects with missing data in plasma endpoints were excluded from the correlation analysis. Results are presented as means ± standard error of the mean (SEM). Two-tailed *p* < 0.05 was considered significant. 

## 3. Results

### 3.1. Subject Demographics, Dietary Assessment and Metabolic Health Characteristics

One hundred and two subjects were screened and 36 men and women, PreDM-IR (*n* = 26) and Reference (*n* = 10), were enrolled (Figure 2). Their dietary patterns and fasting metabolic health indices were characterized. In general, the PreDM-IR group consumed less dietary fiber and less total fruit and vegetables in their usual diet compared to the Reference group (Table 2). The PreDM-IR group had more obesity and higher metabolic and atherogenic risks than the Reference group (Table 3). In a sub-analysis of lean/healthy weight individuals (based on body composition analysis) with PreDM-IR (PreDM-Lean, *n* = 7) or without (Reference-Lean, *n* = 8), fasting glucose and insulin concentrations and HOMA-IR were significantly different (*p* < 0.05).

### 3.2. Gut Microbiome Composition

A total of 229 species (13 phyla) of gut bacteria were identified and 101 species were present in at least five samples at 0.1% relative abundance, mainly belonging to *Firmicutes*, *Bacteroidetes*, *Actinobacteria*, *Proteobacteria* and *Verrucomicrobia* (Appendix A). No significant differences in the *Firmicutes/Bacteroidetes* ratio, within-sample diversity (α-diversity) or between-sample diversity (β-diversity) were found between the PreDM-IR and Reference groups (Appendix A). 

A total of 14 species were differentially abundant (*p* < 0.05) between PreDM-IR and Reference groups (Figure 3). Distinguishing characteristics in PreDM-IR compared to the Reference group included being enriched with the species *Blautia obeum*, *Blautia wexlerae*, *Clostridium clostridioforme* and *Ruminococcus gnavus* and being depleted of *Bacteroides dorei*, *Coprococcus eutactus, Eubacterium eligens* and *Bacteroidetes eggerthii.*

In the sub-analysis of lean/healthy weight individuals, 11 species were differentially abundant (Figure 4). Of these, *B. wexlerae*, *R. gnavus* and *B. eggerthii* were also abundant in the PreDM-IR group as a whole, suggesting these species as possible markers of PreDM independent of obesity. Additionally, *Akkermansia muciniphila* and several *Bacteroides* spp. were depleted in the lean/healthy weight PreDM-IR group, suggesting these microbes are biomarkers of metabolic health.

### 3.3. Predicted and Experimental Analyses of Gut Metabolomics

Functional consequences of gut microbiome compositional shifts were predicted based on metagenomic reads with developed pipelines HUMAnN2 and MelonnPan [25,27] and experimentally evaluated based on the (poly)phenolic metabolome characterized in plasma and urine samples of PreDM-IR and Reference groups after consumption of the RRBtest drink high in distinctive (poly)phenols. 

In the predictive analysis, a total of 14 pathways and eight microbial metabolites were differentially abundant (*p* < 0.05) between PreDM-IR and Reference groups (Appendix A). In general, the PreDM-IR group was characterized with enhanced microbiome functional capacity associated with galactose and stachyose degradation pathways, reduced amino acids and nucleic acid metabolizing pathways and select reduction in predicted metabolomic markers, such as urobilin, hydrocinnamic acid and azelate.

In the experimental analysis, a total of 110 microbial-derived (poly)phenolic metabolites were quantified in plasma and urine samples post-consumption of the RRBtest drink. In general, the PreDM-IR group had significantly lower concentrations of plasma 3,8-dihydroxy-urolithin (urolithin A) derivatives (~54%) and select phenolic acids, such as dihydroxycinnamic acids (~38%), benzoic acids (~33%) and hydroxyhippuric acids derivatives (~37%) compared to the Reference group (*p* < 0.05) (Figure 5). Likewise, the urine samples showed lower concentrations of 3,8-dihydroxy-urolithin derivatives (~52%), phenyl-*γ*-valerolactones (~67%) and select phenolic acids, such as dihydroxycinnamic acids (~42%), benzoic acids (~68%), hydroxyhippuric acids derivatives (~50%) and hippuric acid (~64%) (*p* < 0.05) compared to the Reference group (Figure 6). In a sub-analysis of lean/healthy weight individuals, lower concentrations of plasma 3,8-dihydroxy-urolithin derivatives and dihydroxycinnamic acids were observed in PreDM-Lean compared to Reference-Lean (*p* < 0.05) (Figure 5). 

### 3.4. Putative Associations between Gut Microbiome Composition and Metabolic Health Indices and Microbial (Poly)phenolic Metabolites

Species-level associations with metabolic status and microbial-derived (poly)phenolic metabolites (AUC_0-24h_ post-consumption of the RRBtest drink) are shown in Figure 7. The associations indicate that specific species produce (poly)phenolic metabolites, which may be involved in metabolic regulation/disruption. Eleven bacterial species were positively correlated with glycemic, atherogenic and/or obesity risk factors (*p* < 0.05) (Figure 7). For the species enriched in the PreDM-IR group, *B. obeum* was positively correlated with systolic BP, *B. wexlerae* was positively correlated with fasting glucose and insulin, HOMA-IR, HOMA-β and triglyceride concentrations, *C. clostridioforme* and *R. gnavus* were positively correlated with most glycemic risk indices and whole body and trunk fat mass and percentage (*p* < 0.05). *B. wexlerae*, *C. clostridioforme* and *R. gnavus* were negatively correlated with the AUC_0-24h_ of dihydroxycinnamic acid derivatives (*p* < 0.05). 

A total of 43 species were inversely correlated with glycemic, atherogenic and/obesity risk factors (*p* < 0.05), including most species enriched in the Reference group (Figure 7). A total of 18 gut microbial species were positively correlated with 3,8-dihydroxy-urolithin-derivatives (*p* < 0.05). Out of 10 gut microbial species enriched in the Reference group, four species (*B. dorei*, *C. eutactus*, *Ruminococcus flavefaciens* and *Adlercreutzia equolifaciens*) were positively correlated with 3,8-dihydroxy-urolithin derivatives (*p* < 0.05), but none of the PreDM-IR enriched species were correlated with 3,8-dihydroxy-urolithin (Figure 7). In addition, 15 other gut microbial species were also positively correlated with 3,8-dihydroxy-urolithin derivatives, including *A. muciniphila*, *Bilophila wadsworthia*, *Alistipes* spp. and *Eubacterium* spp. (*p* < 0.05). A total of 10 species were positively correlated with phenyl-*γ*-valerolactones derivatives (*p* < 0.05), but these species were not significantly different between the PreDM-IR and Reference groups. 

## 4. Discussion

This study reported, for the first time, a comprehensive analysis of gut microbiome composition and function in young and middle-aged adults with PreDM-IR. This included analysis of taxa composition and associated function based on predicted and experimental metabolomics in relation to metabolic health/risk indices. The most significant findings in PreDM-IR compared to a metabolically healthy Reference group included: (1) enriched with species *B. obeum*, *B. wexlerae*, *C. clostridioforme* and *R. gnavus* and depleted *B. dorei*, *C. eutactus* and *E. eligens. A. muciniphila* and several *Bacteroides* spp. were depleted in PreDM-IR independent of obesity; (2) depleted predicted and experimental gut microbiome functional capacity associated with (poly)phenol metabolism, such as reduced 3,8-dihydroxy-urolithin, phenyl-*γ*-valerolactones and various phenolic acids; and (3) diabetes risk indices were associated with specific metagenomic shifts and reduced microbial-derived (poly)phenolic metabolites. Collectively, these data indicate structural and functional gut dysbiosis in young and middle-aged adults with PreDM-IR. 

Indices of microbial diversity and an elevated *Firmicutes/Bacteroidetes* ratio have been reported as markers of obesity and metabolic diseases based on 16S rRNA gene sequencing. However, a meta-analysis of four different studies from the Human Microbiome Project and MetaHIT Project revealed that inter-study variability far exceeded differences in composition between lean and obese individuals within each study, indicating that phylum-level taxonomic compositions, e.g., *Firmicutes/Bacteroidetes* ratio, were not simply associated with BMI or obesity [37]. Moreover, previous studies did not show a consistent association between the α- and β-diversities and *Firmicutes/Bacteroidetes* ratio with T2DM or PreDM [4,5,6,7]. One possible explanation is that these studies usually included older adults (60+ years old) and aging-associated gut dysbiosis and loss of diversity may interfere with the interpretation of the results [38,39]. According to our results, no significant differences in α- or β-diversities or *Firmicutes/Bacteroidetes* ratio in the PreDM-IR group relative to the Reference group were identified. However, distinct species-level shifts were apparent between the groups. Ghosh et al. suggested that younger people gain disease-causing microbes, whereas in older cohorts, the loss of beneficial bacteria may associate with disease [39]. Understanding factors such as age that influence disease-/pre-disease-linked changes in the gut microbiome will be critical for designing intervention strategies that promote a healthy gut microbiome and reduce the risk of DM.

The enriched genus *Blautia* and depleted genus *Bacteroides* observed in our PreDM-IR group are in accordance with previous findings in all age groups with T2DM and PreDM [5,6,7]. Depleted butyrate-producing species such as *Faecalibacterium prausnitzii* have been associated with T2DM and PreDM [3,5,6]. However, we did not find a significant shift of *F. prausnitzii* in the PreDM-IR group relative to the Reference group. A meta-analysis of 10 disease–microbiome studies revealed that the effect of “age” on gut microbiome composition was higher than the effect of “T2DM” in these studies and depleted *F. prausnitzii* was associated with elderly groups (60+ years old) [39]. Therefore, aging-associated gut microbiome alterations may interfere with the microbe–disease signature.

*R. gnavus* has been reported previously as uniquely abundant in individuals with Crohn’s disease, a type of inflammatory bowel disease (IBD), and since identified as a “pro-inflammatory” bacterial species contributing to a tendency of gut dysbiosis [10,40]. *R. gnavus* was abundant in our PreDM-IR group and in our sub-analysis of PreDM-IR lean/healthy weight compared to the Reference lean/healthy weight group. Increased gut inflammation and gut permeability have been shown in T2DM patients [41,42]. A five-year prospective study in Korea observed that the incidence of diabetes in patients with IBD was significantly higher compared with non-IBD individuals [43]. Hence, people with PreDM and characterized with abundant *R. gnavus* are likely to be at significantly greater risk of transitioning to T2DM, especially if present with other microbial markers. 

The depletion of *A. muciniphila* has been associated with PreDM and T2DM previously [4,6]. Recent animal and human studies indicate beneficial properties of *A. muciniphila* supplementation on insulin resistance, adiposity and total cholesterol [44,45]. However, another study in adults (20–80 years old) with T2DM reported enrichment in *A. muciniphila* [3]. Decreased *A. muciniphila* has been associated with aging in human and animal models [46,47]. Age-associated alterations in the gut microbiota towards disease configurations may mask a true disease-related microbial signature. Our data reiterate the association of *A. muciniphila* with metabolic health; however, this was not apparent until controlling for obesity in the sub-analysis. 

Urolithins (3,8-dihydroxy-urolithin, 3-hydroxy-urolithin and 3,9-dihydroxy-urolithin: urolithin A, urolithin B and isourolithin A, respectively) are final microbial metabolites of ellagic acids and ellagitannins, which accounted for 30% of total RRB (poly)phenols in the RRBtest drink [31]. *Gordonibacter urolithinfaciens* and *Gordonibacter pamelaeae* have been identified as capable of producing intermediate urolithins (i.e., 3,8,9-trihydroxy-urolithin, urolithin C) [48]. *G. pamelaeae* was detected in 26 subjects in this study at relatively low abundance (<0.1% in 25 subjects). However, the presence and abundance of *G. pamelaeae* had no significant correlation with urolithins in plasma and urine, suggesting other gut microbiota are involved in urolithin metabolism. This study identified 19 gut microbial species positively correlated with urolithin production (*p* < 0.05), among which four species were enriched in the PreDM-IR group (Figure 7). Gut microbiota ellagitannin-metabolizing phenotypes (i.e., urolithins metabotypes (UMs)) have been proposed as metabolic health/disease biomarkers [49]. Three different UMs have been described depending on the type of urolithins produced by microbes: urolithin metabotype A (UM-A) is distinguished by the production of urolithin A, urolithin metabotype B (UM-B) produces urolithin B, isourolithin A and urolithin A, and urolithin metabotype 0 does not produce these final types of urolithins (UM-0) [49,50]. A higher percentage of UM-A was observed in healthy individuals and a higher percentage of UM-B was observed in patients with metabolic syndrome or colorectal cancer [49]. In the present study, among the PreDM-IR group (*n* = 26), 10 qualified as UM-A, 11 as UM-B and 5 as UM-0, who are non-producers. In the Reference group (*n* = 10), 5 qualified as UM-A, 4 as UM-B and 1 as UM-0. An ellagitannin-rich foods study in individuals with metabolic syndrome indicated plasma concentrations of 3,8-dihydroxy-urolithin derivatives were positively correlated with HDL cholesterol and inversely correlated with plasma glucose [51]. Our results indicate lower gut microbial capacity to produce 3,8-dihydroxy-urolithin derivatives in PreDM-IR as a whole and independent of obesity (Figure 5a and Figure 6a). 

Phenyl-*γ*-valerolactones are microbial metabolites of flavan-3-ols, which accounted for 6% of (poly)phenols in the RRBtest drink [31]. The plasma AUC_0–24 h_ of phenyl-*γ*-valerolactones was not significantly different between the PreDM-IR and Reference groups (Figure 5b); however, the PreDM-IR group had significantly lower urinary excretion of phenyl-*γ*-valerolactones compared to the Reference group (Figure 6b). Taken together, these data suggest lower gut microbial capacity to produce phenyl-*γ*-valerolactones in PreDM-IR. Phenyl-*γ*-valerolactones have been associated with anti-inflammatory activity, cardiovascular protective effects and chemo-preventive effects in *in vitro* and *in vivo* studies [52,53]. Gut microbial capacity to produce phenyl-*γ*-valerolactones may be indicative of individuals’ ability to obtain health benefits from usual flavan-3-ols-rich foods, such as tea, fruits, wine and cocoa-derived products. Three putative metabotypes have been proposed according to the urinary excretion of flavan-3-ol metabolites of 11 subjects [54]. These include: metabotype 1 has high excretion of tri- and di-hydroxyphenyl-*γ*-valerolactones while a reduced excretion of 3-(hydroxyphenyl)propionic, metabotype 2 has a medium excretion of dihydroxyphenyl-*γ*-valerolactone while a limited excretion of trihydroxyphenyl-*γ*-valerolactone and 3-(hydroxyphenyl)propionic acid and metabotype 3 has limited production of phenyl-*γ*-valerolactones but a high excretion of 3-(hydroxyphenyl)propionic acid [54]. In our study, 9, 7 and 7 subjects qualified as flavan-3-ol metabotypes 1, 2 and 3, respectively; however, 13 subjects did not meet criteria for the proposed metabotypes. These results suggest further research is required to understand flavan-3-ol metabotypes and their relationship with health/disease status and gut microbial composition. 

Anthocyanins comprise a significant portion (~60%) of RRB (poly)phenols [31]. Unabsorbed anthocyanins are metabolized to various phenolic acids by gut microbiota, contributing to the total phenolic pool. After RRBtest drink consumption, the PreDM-IR group had a lower plasma and urinary AUC_0–24 h_ of dihydroxycinnamic acids, benzoic acids and hydroxyhippuric acids derivatives and lower urinary hippuric acid compared to the Reference group (Figure 5c–f and Figure 6c–f). These data suggest reduced gut microbiota function in catabolizing anthocyanins, which is in accordance with the depletion of several predicted metabolites in the PreDM-IR vs. Reference group (Appendix A). In the sub-analysis of lean/healthy individuals, the PreDM-Lean group had a lower AUC_0–24 h_ of dihydroxycinnamic acids, but not benzoic acids and hydroxyhippuric acids derivatives in plasma compared to the Reference-Lean group. They also had lower concentrations of dihydroxycinnamic acid derivatives, hydroxyhippuric acid derivatives and hippuric acid, but not benzoic acid derivatives in urine. These results suggest that obesity may influence the progression of prediabetes by reducing the gut microbial capacity to metabolize (poly)phenols.

The PreDM-IR group reported consuming less dietary fiber and less total fruit and vegetables intake in their usual diet (Table 2). Diets poor in fruits and vegetables and low in dietary fiber have been associated with increased risk of gastrointestinal diseases, cardiovascular disease and diabetes [55]. Our results support further research on the role of dietary habits, especially dietary fibers, fruits and vegetables, on metabolic health, gut microbiome composition and function.

Shotgun metagenomic analysis enabled in-depth whole genome sequencing to explore richer and more accurate information relative to 16S rRNA amplicon sequencing widely used previously. Young and middle-aged adults (33 ± 11) were a focus of this research to broaden knowledge across age groups and reduce interference of aging-associated gut microbiome alterations. However, the research had limitations. As a pilot and exploratory study, a relatively small number of subjects were enrolled, especially in the Reference group, which may weaken the power to deliver definitive conclusions. That said, few human investigations include a reference control group of similar age, which adds strength to our findings. The gender ratio was imbalanced between the two groups; however, gender was included in the statistical analysis as a covariate to control for any significant effects on endpoints. Plasma and urine samples were collected 0–4 h and 24 h after the RRBtest drink to focus on phenyl-*γ*-valerolactones, urolithins and select phenolic acids. However, collection of urine samples between 24 and 48 h may provide more information on other microbial metabolites, which will be considered in the future.

## 5. Conclusions

Young and middle-aged adults with prediabetes and insulin resistance exhibited shifts in gut microbiome composition and lower microbial capacity to catabolize dietary (poly)phenols compared to metabolically healthy individuals of similar age. Sub-analyses controlling for obesity revealed relationships with specific microbial species that may serve as metagenomic markers of diabetes development and therapeutic targets for cardio-metabolic disease risk reduction. Likewise, comparing our data broadly with other published research supports a need for further research to understand age–diabetes–microbiome associations. Overall, there remains a paucity of knowledge in this area; however, combining metagenomics with other omics approaches while “challenging” the bio-system as demonstrated in the present study will help advance our knowledge at the diet–gut–disease nexus, supporting efforts in precision medicine to prevent diabetes development and improve metabolic health.

## Figures and Tables

**Figure 1 nutrients-12-03595-f001:**
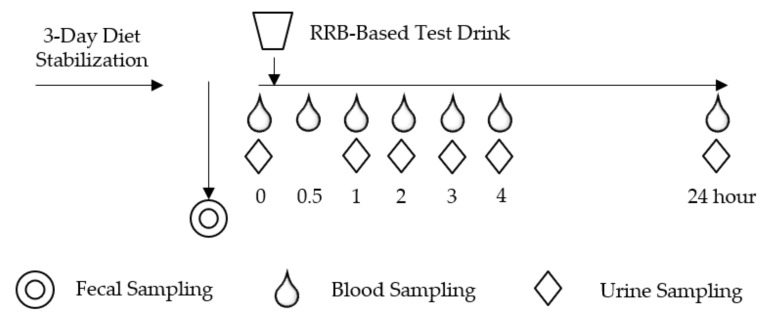
Postprandial study day schema.

**Figure 2 nutrients-12-03595-f002:**
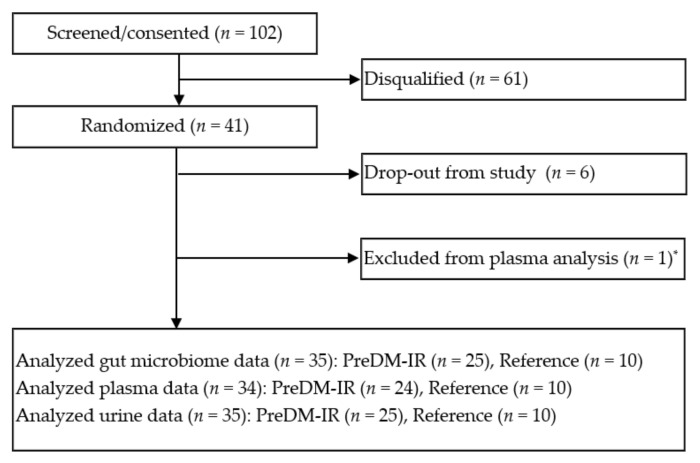
Consolidated Standards of Reporting Trials (CONSORT) flow diagram of the study. * Fail in blood collection.

**Figure 3 nutrients-12-03595-f003:**
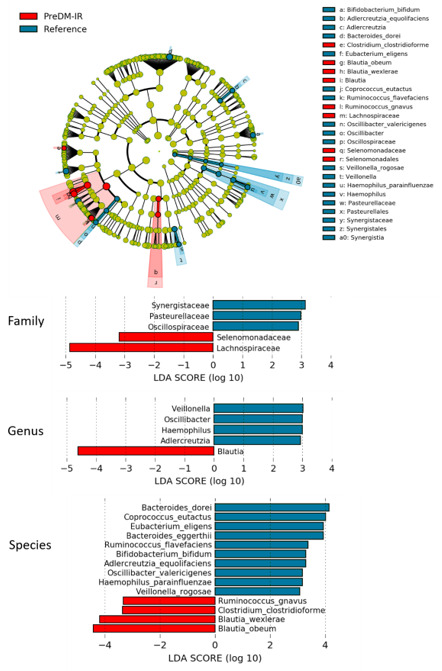
Gut microbiome composition in the PreDM-IR group relative to the Reference group. Linear discriminant analysis (LDA) score (log10) above 2.0 and *p* < 0.05.

**Figure 4 nutrients-12-03595-f004:**
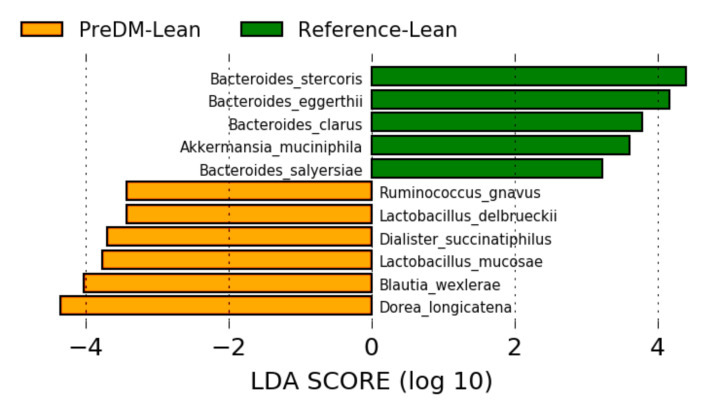
Gut microbiome composition in the PreDM-IR group relative to the Reference group. Linear discriminant analysis (LDA) score (log10) above 2.0 and *p* < 0.05.

**Figure 5 nutrients-12-03595-f005:**
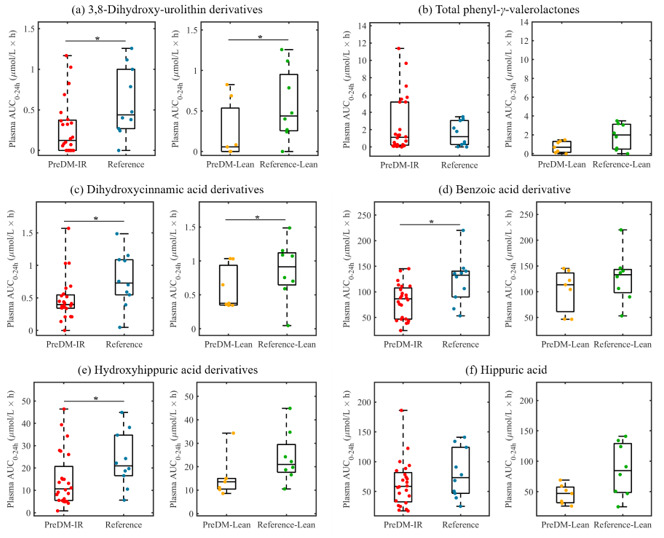
Box plots showing AUC_0–24 h_ (areas under the 24 h curve) of microbial (poly)phenolic metabolites in plasma post-consumption of the red raspberry test (RRBtest) drink in PreDM-IR (red dots) vs. Reference (blue dots) and PreDM-Lean (yellow dots) vs. Reference-Lean (green dots): (**a**) 3,8-dihydroxy-urolithin derivatives; (**b**) total phenyl-*γ*-valerolactones; (**c**) dihydroxycinnamic acids derivatives; (**d**) benzoic acids derivatives; (**e**) hydroxyhippuric acids derivatives; (**f**) hippuric acid. * Significantly different, *p* < 0.05.

**Figure 6 nutrients-12-03595-f006:**
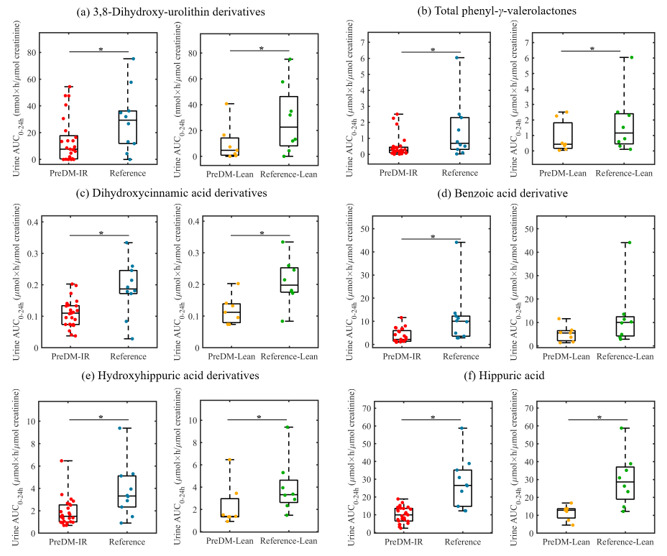
Box plots showing AUC_0–24 h_ (areas under the 24 h curve) of microbial (poly)phenolic metabolites in urine post-consumption of the RRBtest drink in PreDM-IR (red dots) vs. Reference (blue dots) and PreDM-Lean (yellow dots) vs. Reference-Lean (green dots): (**a**) 3,8-dihydroxy-urolithin derivatives; (**b**) total phenyl-*γ*-valerolactones; (**c**) dihydroxycinnamic acids derivatives; (**d**) benzoic acids derivatives; (**e**) hydroxyhippuric acids derivatives; (**f**) hippuric acid. * Significantly different, *p* < 0.05.

**Figure 7 nutrients-12-03595-f007:**
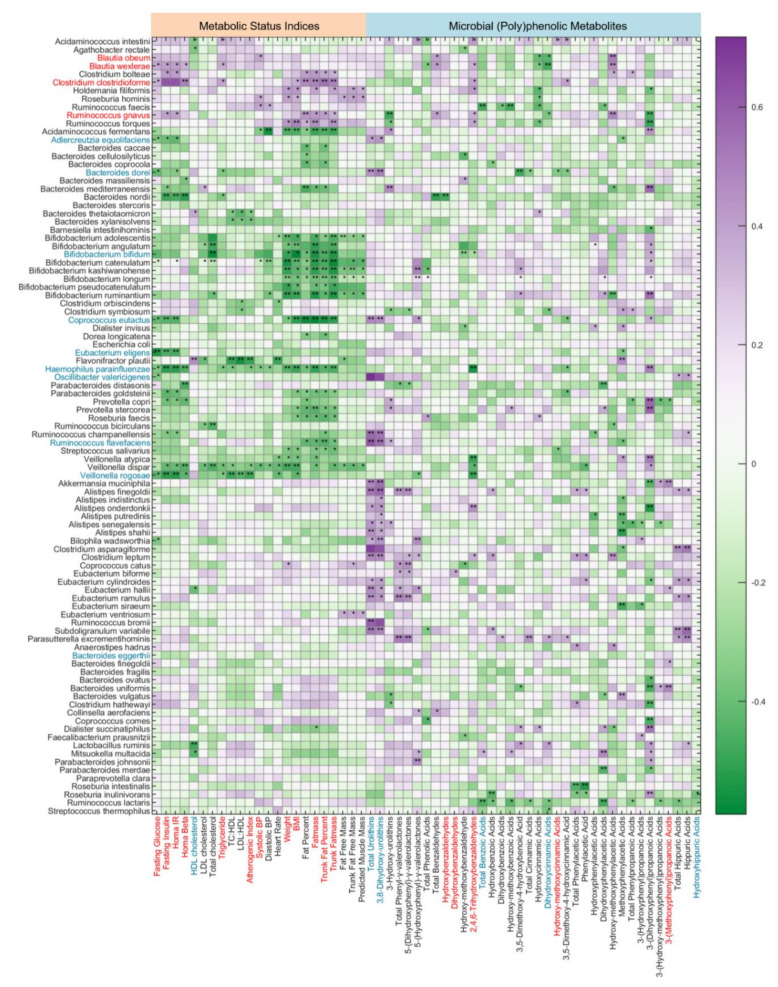
Putative gut microbiome composition associations with metabolic risk indices and microbial (poly)phenolic metabolites. Biomarkers with statistically significant correlation in the PreDM-IR and the Reference groups are colored red and blue, respectively. Heatmap color intensity represents the magnitude of correlation. Purple = positive correlations; green = negative correlations. Significant correlation, * *p* < 0.05 and ** *p* < 0.01.

**Table 1 nutrients-12-03595-t001:** Nutrient composition of red raspberry-based test drink (RRBtest) ^1^.

Item Name	RRBtest Drink
Quantity (g)	414
Energy (kcal)	361
Protein (g)	2.1
Fat (g)	1.4
Carbohydrate (g)	85
Fiber (g)	8
Sugar (g)	75
RRB sugar	10
Added sugar	65
Total (poly)phenols (mg)	388
Anthocyanin (mg)	237
Ellagitannin/ellagic acid (mg)	125

^1^ Protein, carbohydrate, fiber and sugar contents derived from USDA Food Composition Databases. Total (poly)phenols, anthocyanin and ellagitannin/ellagic acid contents were quantified using the method described in Zhang et al. [12].

**Table 2 nutrients-12-03595-t002:** Dietary intake assessment of PreDM-IR (prediabetes and insulin-resistance) and Reference groups ^1^.

Nutrient and Food Group	PreDM-IR (*n* = 26)	Reference (*n* = 10)	*p* Value
Energy (kcal)	1917 ± 142	1832 ± 193	NS
Protein (g)	86 ± 6	93 ± 14	NS
Total fat (g)	80 ± 7	62 ± 8	NS
Carbohydrate (g)	213 ± 18	233 ± 22	NS
Sugars, total (g)	84 ± 10	70 ± 7	NS
Added sugar (g)	12 ± 2	6 ± 1	NS
Fiber, total dietary (g)	16 ± 2	24 ± 4	0.02
Total fruits (cup)	0.8 ± 0.2	1.0 ± 0.3	NS
Total vegetable (cup)	1.6 ± 0.2	2.0 ± 0.5	NS
Total fruit and vegetable (cup)	2.3 ± 0.3	3.0 ± 0.5	0.05

^1^ Data were collected by the Automated Self-Administered 24-h (ASA24) Dietary Assessment Tool.

**Table 3 nutrients-12-03595-t003:** Subject demographic and metabolic health characteristics ^1^.

Metabolic Health Indices	PreDM-IR (*n* = 26)	Reference (*n* = 10)	*p*-Value ^2^	PreDM-Lean (*n* = 7) ^3^	Reference-Lean (*n* = 8) ^3^	*p*-Value ^4^
Age (years)	34 ± 11	31 ± 9	NS	28 ± 12	30.5 ± 10	NS
Female: Male	12:14	7:3	NS	2:5	6:2	NS
CAU/AA/AS/HIS	8:6:9:3	3:2:3:2	NS	1:1:5:0	2:1:3:2	NS
Fasting glucose (mmol/L)	5.8 ± 0.1	4.9 ± 0.1	<0.0001	5.6 ± 0.2	4.9 ± 0.1	0.008
Fasting insulin (pmol/L)	88.2 ± 8.7	34.0 ± 4.1	<0.0001	67.0 ± 6.8	35.0 ± 4.5	0.01
HOMA-IR	3.3 ± 0.3	1.1 ± 0.1	<0.0001	2.4 ± 0.3	1.1 ± 0.1	0.004
HOMA-β%	115.5 ± 11.3	70.1 ± 7.9	0.01	91.6 ± 9.2	72.5 ± 9.8	NS
TC (mmol/L)	4.7 ± 0.2	4.3 ± 0.2	NS	4.4 ± 0.1	4.4 ± 0.3	NS
HDL-C (mmol/L)	1.3 ± 0.1	1.5 ± 0.1	NS	1.3 ± 0.1	1.5 ± 0.2	NS
LDL-C (mmol/L)	2.9 ± 0.1	2.5 ± 0.1	NS	2.7 ± 0.1	2.5 ± 0.2	NS
TG (mmol/L)	1.1 ± 0.1	0.7 ± 0.1	0.03	0.9 ± 0.1	0.7 ± 0.1	NS
TC/HDL-C	3.7 ± 0.2	3.0 ± 0.2	NS	3.6 ± 0.4	3.0 ± 0.2	NS
LDL-C/HDL-C	2.3 ± 0.1	1.8 ± 0.1	NS	2.3 ± 0.3	1.8 ± 0.2	NS
Atherogenic Index	2.7 ± 0.2	2.0 ± 0.2	0.04	2.6 ± 0.4	2.0 ± 0.2	NS
Systolic BP (mmHg)	119.3 ± 2.1	110.0 ± 4.0	0.07	116.3 ± 3.2	106.3 ± 3.9	NS
Diastolic BP (mmHg)	74.9 ± 1.5	68.4 ± 3.5	0.08	70.5 ± 2.2	64.9 ± 2.9	NS
Heart rate (beats per minute)	71.6 ± 1.7	65.4 ± 2.6	NS	69.4 ± 3.0	66.6 ± 2.9	NS
Weight (kg)	84.2 ± 4.3	64.3 ± 4.9	0.005	66.4 ± 4.2	57.7 ± 2.8	NS
BMI (kg·m^2^)	28.7 ± 1.2	22.5 ± 1.2	0.001	22.8 ± 0.9	20.9 ± 0.7	NS
Waist (cm)	95.2 ± 3.1	78.8 ± 4.4	0.002	81.6 ± 4.3	73.2 ± 2.9	NS
Whole body fat%	30.1 ± 1.9	24.7 ± 3.1	NS	19.4 ± 2.2	22.3 ± 2.8	NS
Whole body fat mass (kg)	26.0 ± 2.3	16.3 ± 3	0.007	12.9 ± 1.7	12.9 ± 1.8	NS
Fat-free mass (kg)	57.9 ± 2.9	47.6 ± 3.3	NS	53.1 ± 3.2	44.6 ± 2.7	NS
Trunk fat%	30.4 ± 1.8	23.3 ± 3.1	0.05	19.4 ± 1.9	20.3 ± 2.6	NS
Trunk fat mass (kg)	14.4 ± 1.3	8.7 ± 1.8	0.02	7.2 ± 1.1	6.5 ± 1.0	NS
Trunk fat-free mass (kg)	31.3 ± 1.4	26.2 ± 1.5	NS	28.9 ± 1.7	24.6 ± 1.1	NS
Predicted muscle mass (kg)	29.7 ± 1.2	25.1 ± 1.5	NS	27.8 ± 1.7	23.6 ± 1.1	NS

^1^ Mean ± SEM for continuous variables (except age, mean ± SD). NS, non-significant. CAU, Caucasian; AA, African American; AS, Asian; HIS, Hispanics and Latino; HOMA-IR, homeostasis model assessment of insulin resistance; HOMA-β, homeostasis model assessment of β cell function, total cholesterol; HDL-C, high-density lipoprotein cholesterol; LDL-C, low-density lipoprotein cholesterol; TG, triglyceride; BP, blood pressure; BMI, body mass index. ^2^ Comparison between PreDM-IR and Reference groups adjusted for gender. ^3^ Lean (healthy weight) was defined according to the Tanita body fat percentage chart [36]. ^4^ Comparison between PreDM-Lean and Reference-Lean subgroups adjusted for gender.

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
