# Peer review of "Functional Deficits in Gut Microbiome of Young and Middle-Aged Adults with Prediabetes Apparent in Metabolizing Bioactive (Poly)phenols"

_nutrients, 2020, doi:10.3390/nu12113595_

Round 1

Reviewer 1 Report

In this manuscript Zhang et al. have analyzed gut microbiome composition and its related functional capacity to metabolize fruit (poly)phenols in healthy and prediabetic-insulin resistant individuals. To this end, they used shotgun sequencing. They concluded that young and middle-aged adults with prediabetes and insulin resistance exhibited shifts in gut microbiome composition and lower microbial capacity to catabolize dietary (poly)phenols compared to metabolically healthy individuals of similar age.

Comments to authors

The manuscript is clearly presented and conclusions are based on experimental data. Novelty of the work is partial. The major flaws of the manuscript are the low number of samples (it may not be extrapolated to a bigger population) and that it is descriptive, without attempt to demonstrate cause-effect relationship.

Author Response

Response to Reviewer 1 Comments

In this manuscript Zhang et al. have analyzed gut microbiome composition and its related functional capacity to metabolize fruit (poly)phenols in healthy and prediabetic-insulin resistant individuals. To this end, they used shotgun sequencing. They concluded that young and middle-aged adults with prediabetes and insulin resistance exhibited shifts in gut microbiome composition and lower microbial capacity to catabolize dietary (poly)phenols compared to metabolically healthy individuals of similar age.

Comments to authors

The manuscript is clearly presented and conclusions are based on experimental data. Novelty of the work is partial. The major flaws of the manuscript are the low number of samples (it may not be extrapolated to a bigger population) and that it is descriptive, without attempt to demonstrate cause-effect relationship.

Response: Thank you for pointing out the limitation of the sample size. Indeed, the sample size is limited by the available funding. In this exploratory study, we chose shotgun whole genome sequencing to obtain high taxonomic resolution (species-strains), high coverage, functional profiling and metabolomics prediction, which allow the association with (poly)phenolic metabolites. However, shotgun sequencing is much more expansive than widely used 16S sequencing and required more input in bioinformatic analysis. We would like to include more subjects and try to explore cause-effect relationship in the future using the methods developed and validated in this study.

Reviewer 2 Report

The present study aims at investigating the association between microbiota composition and (poly)phenolic metabolites after raspberry consumption. As the authors mentioned, the novelty of the work is the age of the population, being the participants of the study young prediabetes-insulin resistant subjects. The work is well written but some points need to be implemented.

MAJOR COMMENTS

The authors quantified the main (poly)phenolic compounds they detected in urine and plasma samples. However, they did not specify how they quantify those metabolites, nor which metabolites they identified. Moreover, the authors collected urine 24 h after raspberry consumption, but an important fraction of colonic metabolites are mainly generated within 24-48h.

The authors reported the association results for PreDM-IR vs Reference group and for PreDM-lean and reference-lean group. However, no information about these two sub-groups are reported. Were they homogeneous? How many people did they include? Which were the characteristics?

MINOR COMMENTS

Introduction, Line 35: impaired glucose tolerance (IGT) has been already specified at line 34.

M&M, line 94: “After a low-(poly)phenol diet stabilization period”: how long was that period? This is an important aspect to specify since the microbial degradation of native (poly)phenolic compound generally takes more than 36h.

Line 157 and 162: The authors did not report the provider of the HPLC columns

RESULTS: Considering the dietary assessment of the groups (table 2), could a different intake of fiber have partially influenced the results?

DISCUSSION: lines 360-366: Gordonibacter urolithinfaciens and Gordonibacter pamelaeae have been proposed as species responsible for urolithin production. Did the authors check for these specific species?

Lines372-381: The authors could mention the metabotypes recently proposed for valerolactone production.

The name of the (poly)phenolic metabolites could be aligned with the new work on nomenclature recently published (Kay et al. Recommendations for standardizing nomenclature for dietary (poly)phenol catabolites. Am J Clin Nutr. 2020 Oct 1;112(4):1051-1068)

Author Response

Response to Reviewer 2 Comments

Thank you for your review, comments and suggestions. The point-by-point responses are listed below.

Point 1: The authors quantified the main (poly)phenolic compounds they detected in urine and plasma samples. However, they did not specify how they quantify those metabolites, nor which metabolites they identified.

Response 1: Thank you for important suggestions. The quantification method is described in detail in our recently published paper (https://doi.org/10.3390/molecules25204777), including available standards, method validation (linearity, limit of detection, limit of quantification, precision, recovery, matrix effect), and MRM transitions (Revised at Line 168-173). Different metabolites were quantified individually and grouped based on their structure. The major (poly)phenolic groups significantly different between PreDM-IR and Reference groups were presented in Figures 5 and 6.

Point 2: Moreover, the authors collected urine 24 h after raspberry consumption, but an important fraction of colonic metabolites are mainly generated within 24-48h.

Response 2: Thank you for important suggestions. When we designed this study, we did a two-subject pilot trial and we observed the major anthocyanin catabolites, cinnamic acids and benzaldehydes peaked before 24 h and the 24 h urolithin concentration was higher than 48 h. Thus, we only designed one next-morning follow-up. However, we agree that for a more diverse microbial metabolites pool 24-48 h urine samples may disclose more information. We will consider collecting 24-48h in our future studies (Line 442-445)

Point 3: The authors reported the association results for PreDM-IR vs Reference group and for PreDM-lean and reference-lean group. However, no information about these two sub-groups are reported. Were they homogeneous? How many people did they include? Which were the characteristics?

Response 3: Thank you for important suggestions. The demographic and metabolic health characteristic information of subgroups are added to Table 3.

MINOR COMMENTS

Point 4: Introduction, Line 35: impaired glucose tolerance (IGT) has been already specified at line 34.

Response 4: Thanks. Revised accordingly at Line 36.

Point 5: M&M, line 94: “After a low-(poly)phenol diet stabilization period”: how long was that period? This is an important aspect to specify since the microbial degradation of native (poly)phenolic compound generally takes more than 36h.

Response 5: Thanks for pointing it out. The low-(poly)phenol diet stabilization period is 3 days. Diet stabilization time is added at Line 95.

Point 6: Line 157 and 162: The authors did not report the provider of the HPLC columns

Response 6: Thanks for pointing it out. The provider information is added at line 159 and line 164.

Point 7: RESULTS: Considering the dietary assessment of the groups (table 2), could a different intake of fiber have partially influenced the results?

Response 7: Thanks for pointing it out. Yes. Habitual diet low in fibers may influence metabolic health and gut function.  A discussion paragraph is added at line 429-433.

Point 8: DISCUSSION: lines 360-366: Gordonibacter urolithinfaciens and Gordonibacter pamelaeae have been proposed as species responsible for urolithin production. Did the authors check for these specific species?

Response 8: Thanks for pointing it out. Gordonibacter urolithinfaciens was not detected. Gordonibacter pamelaeae was identified in 26 subjects, however, the relative abundance was low (< 0.1% in 25 subjects) and no significant correlation with total or specific urolithin production. The discussion of Gordonibacter was added to line 375-381.

Point 9:  Lines372-381: The authors could mention the metabotypes recently proposed for valerolactone production.

Response 9: Thanks for pointing it out. The discussion of flava-3-ol metabotypes is added to line 406-415.

Point 10:  The name of the (poly)phenolic metabolites could be aligned with the new work on nomenclature recently published (Kay et al. Recommendations for standardizing nomenclature for dietary (poly)phenol catabolites. Am J Clin Nutr. 2020 Oct 1;112(4):1051-1068)

Response 10: Thanks for helpful suggestion. We aligned the nomenclature accordingly in the manuscript (Revised at Line 173 and Figures 5-7)

Round 2

Reviewer 2 Report

Authors revised the paper as suggested and the paper can be accepted in the present form.